# Construction of an Integrated Framework for Complex Product Design Manufacturing and Service Based on Reliability Data

**Jinfeng Wang** [1,2], **Zhan Meng** [1], **Dapeng Gao** [3] **and Lijie Feng** [1,2,*]

1   School of Management, Zhengzhou University, Zhengzhou 450001, China; jinfengwang@zzu.edu.cn (J.W.); 202011301010333@gs.zzu.edu.cn (Z.M.)
2   China Institute of FTZ Supply Chain, Shanghai Maritime University, Shanghai 201306, China
3   Logistics Management Section, SAIC Motor Passenger Vehicle Company (Zhengzhou Branch), Zhengzhou 450048, China; gaodapeng@saicmotor.com
*   Correspondence: ljfeng@shmtu.edu.cn

**Abstract:** With the application of new-generation information technology in the full life cycle process of a complex product, it is showing the characteristics of multi-source, real-time, heterogeneous, cross-domain transmission. Large data volume and low value density emerge in the process of complex product design manufacturing and services (DMS). This leads to "information islands" and insufficient utilization of cross-domain reliability data in the process of integration of DMS for complex product R&D design data, manufacturing data and operation and maintenance services (O&MS) data. This paper proposes and illustrates a framework of complex product DMS integration based on reliability data, including complex product design optimization based on manufacturing and service reliability data, complex product intelligent manufacturing process optimization based on real-time reliability data and complex product O&MS optimization based on multi-source heterogeneous reliability data. Additionally, it then realizes complex product design reliability and optimization, manufacturing process reliability and optimization and O&MS reliability and intelligent decision optimization based on reliability data. Finally, the DMS integration framework based on reliability-data-driven proposal is corrected through the case of engine MDS integration, which can effectively improve the cross-domain reliability data utilization and overall product reliability of complex products. The proposed framework extends the application of reliability theory in the process of complex product DMS integration and provides a reference for enterprises in the R&D, manufacturing and O&MS of complex products.

**Keywords:** complex product; integration of design manufacturing and service; reliability data; engine





## 1. Introduction

With the widespread application of smart manufacturing, Internet of Things(IoT), cloud computing, 5G and other new-generation information technologies, the traditional manufacturing model is undergoing radical changes, and the integration of complex product DMS [1] has become the main direction for the development of advanced manufacturing in various countries, which puts forward higher requirements for the acquisition, storage, analysis, visualization and intelligent decision support of the full life cycle (FLC) process data of complex product design, manufacturing and services. The deep integration of digitization, informatization and intelligence makes the products more and more powerful, more and more perfect, more and more complex and more intelligent, and the problem of cross-domain reliability in the integration of design, manufacturing and service of complex products (e.g., new energy vehicles, high-speed rail, aerospace equipment, high-end machine tools and engines, etc.) is particularly prominent. In 1957, the Electronic Equipment Reliability Advisory Group of the U.S. Department of Defense gave the definition, indices and evaluation methods of reliability in the report "Reliability of Military Electronic Equipment", which laid the foundation for the development of reliability theory in the

fields of complex equipment [2], machinery [3], software [4], manufacturing [5], design [6] and PLC management [7], and it provided support for the development of product DMS integration in the fields of aerospace equipment, new energy vehicles, high-end machine tool equipment and engines, and produced great economic benefits.

Reliability data are an important indicator to measure the quality, function, safety and maintenance of complex product design, manufacturing and service. How to effectively obtain the reliability data of complex products in the design, manufacturing and service stages? How to accurately analyze the intrinsic factors and correlations that affect the optimization of complex product design, manufacturing performance improvement and service quality improvement through the trend of reliability data? How to utilize the cross-domain reliability data from complex product design in the manufacturing stage, complex manufacturing process and operation and maintenance service process, realize the closed-loop feedback mechanism of design, manufacturing and service life cycle, and effectively promote the wide application of reliability data in the FLC of complex product DMS is the main problem faced by the academia and industry.

In order to promote an effective solution to the above problems, countries have introduced a series of top-level strategic support policies to enhance the global competitiveness of complex products. Germany proposed the "Industry 4.0" development strategy, which aims to realize the integration of intelligent production and service of industrial products through the integration of information network and physical production system. USA has put forward the concept of "Industrial Internet", taking the advantages of information technology to connect people, data and machines to achieve a high degree of integration of global industrial systems with advanced computing, analysis, sensor technology and the Internet and efficient use of industrial big data. Japan proposed a "new robot strategy" based on the application of technologies such as data terminalization, networking and cloud computing. The United Kingdom promoted the integration of remanufacturing and services centered on production to achieve a rapid and sharp response to consumer demand "UK Manufacturing 2050" plan. China has focused on new energy vehicles, major equipment manufacturing, new sensors and other key industrial complex product areas, such as design, manufacturing and services, and other key links to carry out the integration of new generation information technology and manufacturing equipment engineering applications to implement the "Made in China 2025" strategy. In essence, the implementation of the above strategies is inseparable from the data collection, analysis and innovative utilization of the whole process of complex product design, manufacturing and service.

The new generation of information technology (e.g., high-precision sensors, IoT, 5G and DT, etc.) has been widely used in the process of complex product design, manufacture and service. It realizes the efficient collection, analysis and application of a large amount of process data and product structure data generated in the process of complex product design and trial production, product quality status and various equipment status data generated in the manufacturing process, as well as operating status and various performance operation and maintenance and parameter data generated during the service process. Although large amounts of multidimensional, real-time structured and unstructured data support the implementation of complex product life management (PLM), it is difficult to use a single model to achieve integrated management of complex product DMS driven by industrial big data. Therefore, the data sharing and cross-domain collaboration model based on reliability data to establish multi-granularity, multi-stage and multi-level integration of complex product design, manufacturing and service has become a new direction for the PLM of complex products.

Based on the above, this paper proposes a cross-domain integrated framework of reliability data in the integration of complex product DMS with the premise that new-generation information technologies, such as high-precision sensors, IoT, digital twin (DT) and 5G, are widely used in the integration of complex product DMS. This is to promote the reliability design optimization of complex products, the reliability and optimization of multi-agent manufacturing processes and the improvement of intelligent operation and

maintenance and service reliability, and to provide theoretical and practical support for the high-quality development of strategies, such as "Industry 4.0", "Made in China 2025", etc.

## 2. Related Works

Based on the understanding of the integration of reliability data and DMS, domestic and foreign scholars have conducted research on complex product design optimization, manufacturing process optimization and intelligent decision making of O&MS.

The rapid development of new-generation information technology has led to the further development of big data acquisition, processing, analysis and application in the full life cycle (FLC) management process of complex product design, manufacturing and service, among which the more typical ones are based on big-data-driven [8], IoT [9], Industrial Internet [10], DT [11], decision theory and group decision making [12], etc. For example, big-data-driven complex product life cycle management emphasizes the characteristics of big data through the integrated application of data and knowledge at all stages of the life cycle to enhance the intelligent decision-making capability of complex product life cycle management. IoT focuses on its connotation in the FLC quality management of complex products and further drives the change of enterprise management model, manufacturing model, service model and business model. The Industrial Internet platform aims to provide real-time management of data and information by accessing, managing and controlling product-related data, information and knowledge at all stages of the life cycle, including beginning of life (BOL), middle of life (MOL) and end of life (EOL). DT provides support for PLM customizability, multi-level collaboration, full life cycle data management, knowledge sharing and reuse, and digital simulation. The concept of decision theory and group decision support avoids research deficiencies in PLM and effectively supports industrial organizations in managing their products and related data in all phases of the product life cycle, improving the ability of organizations to manage product development activities, cross-organizational functions and business units, and collaboration between organizations.

More and more studies show that complex products gradually show design for manufacturing, manufacturing services and service reversal for complex product design optimization and manufacturing process optimization. Zhang Q et al. (2019) [13] proposed the concept and architecture of smart connected products to systematically review current research on new paradigms for product development, manufacturing and services, providing a new approach for the study of smart development, smart manufacturing and data-driven services for complex product life cycle management in a smart connected environment.

### 2.1. Research on Reliability of Complex Product Design

The reliability of complex products is largely influenced by the decisions made in the design process. Design defects are transmitted exponentially into the manufacturing and service processes, and design errors or changes are increasingly costly. Therefore, the scientific design method is adopted to discover design defects in time and correct deviations at the beginning of the design, which has become the direction of the reliability design of complex products.

Costa N et al. [14] introduced service design to manufacturing enterprises, combining the human-oriented service design perspective with the organizational network-oriented "product-service" system perspective to form a new production service system (PSS) method. The new PSS approach has improved the original product design model and method, and it has been applied in the manufacturing industry and has effectively promoted the combination of PSS and service design. Chen RR et al. [15] proposed a model validation method implemented in the design phase, that is, the comparison of explicit product requirements with system design properties in systems modeling language (SysML) through the relation-based modeling for static properties (RMSP) method, which enables designers to detect product defects earlier, thus reducing the cost of modifications in the manufacturing and service phases. Sassanelli C et al. [16] proposed the design for X (DfX) approach for parallel engineering, so as to realize the information sharing between

the design and service phases, and systematically support the PSS design process. Limon S et al. [17] considered accelerated degradation testing (ADT) as an effective means to achieve rapid assessment of product reliability during the product design and development phase and provide complex modeling of ADT with multiple acceleration factors. Zhang N et al. [18] constructed a supernetwork model for complex product design through identification and analysis of design elements, sub-network and supernetwork model construction and supernetwork performance analysis, and they took the design process of wind turbines as an example, indicating that the supernetwork model can provide more process information for the collaborative design of wind turbines and can effectively improve the efficiency of collaborative design of wind turbines. Zhou JG et al. [19] proposed mathematical concepts of green design methods (including domain redefinition, failure rate and reliability expression, etc.) based on the life cycle management process, which have implications for electrical product reliability enhancement and resource utilization enhancement. Xue DY et al. [20] proposed a new integrated framework for the optimal design of complex mechanical products by proposing hybrid schemes, developing hybrid simulation methods and hybrid optimization methods, considering modeling, simulation and optimization.

With the continuous improvement of the functional requirements of personalized products, the product structure is becoming more and more complex. The design of a complex engineering product often involves cross-domain collaboration and integration, such as machinery, electricity and automation. However, traditional design methods lack multidisciplinary coordination, leading to barriers to interaction between design stages and a disconnect between product design and prototyping. Tao F et al. [21] proposed the concept of digital twin-driven intelligent design, which pioneered the way in which digital twin technology can improve the design process of different types of products. Qi QL et al. [22] proposed a five-dimensional digital twin model, which provides a technical and basis reference for the application of digital twins in complex product design. Wu Y et al. [23] proposed a new digital twin-enabled multidisciplinary collaborative design method through multidisciplinary knowledge collaboration, multidisciplinary collaborative modeling and multidisciplinary collaborative simulation.

### 2.2. Research on Reliability of Complex Product Manufacturing

With the introduction and application of new information technology in the manufacturing industry, various advanced manufacturing models and national strategies have been proposed and received more and more attention, such as Industry 4.0, Industrial Internet, Cyber-Physical Systems or Network Manufacturing, China Manufacturing 2025, Internet + Manufacturing, Cloud Manufacturing (CMfg), etc.

Combined with emerging technologies, such as cloud computing, IoT, service-oriented technology and high-performance computing, Wang TR et al. [24] proposed a new manufacturing paradigm, CMfg, to solve the bottleneck of informatization development and manufacturing applications. CMfg is becoming an advanced service-oriented manufacturing model for manufacturing. Yuan MH et al. [25] combined time, composability, quality, availability, reliability and cost, proposed a CMfg service quality evaluation index system and constructed a new service quality evaluation model by combining service reliability and credibility, combination complexity and synergy with execution cost, which is important for promoting high-quality service management in manufacturing industries. Kusiak A [26] argued that the future manufacturing enterprise will be highly digitalized, and the traditional design-for-dedicated manufacturing concept will be transformed into design-for-open manufacturing. In many cases, the manufacturing process will become a manufacturing as a service system, that is, service-oriented manufacturing. He YH et al. [27] proposed a fuzzy polymorphic manufacturing system task reliability assessment method based on extended stochastic flow network (ESFN) and validated the proposed method with a manufacturing system producing ferrite phase shifting units as an example. Xu JZ et al. [28] established an evolutionary game model between the government and CMfg companies based on the

characteristics of complex network structures using game theory methods. The research results show that green synergy benefits, cloud platform supervision, government rewards and punishments, etc., can promote the cooperative relationship between CMfg enterprises, help promote the collaborative green innovation of CMfg enterprises and enhance their core competencies.

In addition, the need for excellent complex product reliability and an overall trend toward smart and connected manufacturing systems have given rise to digital twin manufacturing. Tao F. et al. [29] showed how DT has become the core of the intelligent manufacturing process and analyzed how to use DT to improve the efficiency of intelligent manufacturing and how to combine DT with new technologies to achieve efficient and intelligent manufacturing. Yi Y. et al. [30] proposed a DT reference model for intelligent assembly process design and proposed a three-layer intelligent assembly application framework based on DT. Touckia JK. et al. [31] believe that people's demand for sustainable manufacturing and customization products is increasing—and DT have been widely used in intelligent manufacturing—and propose a reconfiguration of DT design and simulation for manufacturing systems model.

### 2.3. Research on Reliability of Complex Product Service

Guided by the technological revolution, widely discussed paradigms, such as servitization are gradually pushing manufacturers to provide more and more complex solutions [32], and the development of industry has put forward higher demands for reducing manufacturing operating costs and improving service reliability, etc. Complex products involve many professional fields, such as machinery, electronics, hydraulics and computers, and they require regular and irregular maintenance to prolong their service life, thereby increasing the cost of the product during its life cycle. Therefore, for manufacturers of complex mechanical products, it is very important to achieve the best match between reliability and economy. If one wants to survive in the fierce market competition, one must compete with other similar products on the market, clearly understand the functions and performance of one's own products, find out the gaps and improve them.

Pang JH et al. [33] developed an intelligent product quality analysis and management system based on rough set (RS) and analytic hierarchy process (AHP), and the results showed that a data-driven condition monitoring and quality analysis system is an important tool for preventing disasters in complex electromechanical products. Wang YR et al. [34] proposed an integrated solution for complex product manufacturing operation and maintenance based on DT for the lack of deep integration between manufacturing operation and maintenance and information island in manufacturing and operation and maintenance links, and they demonstrated the feasibility and effectiveness of the integrated approach by combining the bogie failure prediction of a model of rolling stock. Chang FT et al. [35] believed that the product service system has become a hot topic in the current complex product management field, and the deep integration of products and services has become the key to enhancing product market competitiveness, especially in the service stage. The integration of high-end manufacturing equipment (HEME) and maintenance-repair-overhaul (MRO) services is still not enough to guarantee product functional availability. Based on the perspective of complex network, an integrated product service network model based on functional availability is constructed. This model not only realizes the integration of products and services, guarantees related functions but also provides an important foundation for the identification of key nodes and effective service configuration for availability. Rath N et al. [36] believed that the deterioration of aero-engine performance has a great impact on the reliability, availability and life cycle of complex products, developed an engine health detection system and carried out research on engine health monitoring, diagnosis and prediction technology. The results show that acquiring, analyzing and utilizing engine health information is a must for condition-based maintenance, which ensures high availability of complex products and reduces downtime and operating costs.

In addition, Tao Fei et al. [37] analyzed the connotation of service, product service and manufacturing service, combined DT with service and proposed a general DT to enhance product optimization services, such as DT-driven fault diagnosis methods, DT-driven prediction and health management, etc. Zhuang CB. et al. [38] proposed an intelligent production management and control method framework based on DT, which realized the DT and big-data-driven prediction of the assembly workshop and the production management and control service of the assembly workshop based on the DT. Hu ZT. et al. [39] studied the problems of DT in the manufacturing process of large and complex products and proposed a manufacturing service platform for technical DT to improve the quality and efficiency of marine diesel engine production.

*2.4. Research on Integrated Reliability of Complex Product Design Manufacturing and Services*

Complex products have complex structures, long R&D cycles, many manufacturing links and complex operation and maintenance. Based on the product life cycle perspective, research on the integration of complex product design, manufacturing and service can significantly improve the R&D progress and greatly reduce R&D costs. The integrated reliability system fits the characteristics of the development of complex products in the whole life cycle, which can realize the digitization of reliability work in the whole development cycle, improve the comprehensiveness, flexibility and convenience of reliability work and the efficiency of R&D and manufacturing of complex products.

The deep integration of design, manufacturing and service is the key to realizing the core competitiveness of enterprises with complex products. Lyu Z et al. [40] designed an integrated reliability system for spacecraft development, solving key issues, such as spacecraft structural safety and design optimization, which provided a unified working platform for product design, manufacturing and reliability work and realized the application of spacecraft reliability in the entire development cycle. The shift from product ownership to integrated solutions PSS is expected to lead to a higher customer satisfaction in many cases compared to providing products and services separately. Fargnoli M et al. [41] proposed a service that supports manufacturers in designing their products in relation to the normal function of their products, an approach based on the synergistic use of PSS quality function deployment, axiomatic design and service blueprinting tools to reduce over-engineering risk and product design conflict, providing customers with high-reliability products. Wang YH et al. [42] synthesized the current field of TRIZ service design knowledge system and the emerging field of non-TRIZ service design knowledge system and proposed a design-oriented system emergence thinking method. Violeta DB et al. [43] developed a smart manufacturing DT demonstrator, enabling cross-domain interoperability among DMS and reducing implementation costs. Manufacturing firms are increasingly inclined to offer a combination of products and services to consumers. Bertoni A et al. [44] proposed a model-based approach to evaluate the life cycle cost of a PSS that is already in the conceptual design stage using methods such as CAE, and a case study of the development of a commercial jet engine turbine candidate structure was carried out.

Further, many scholars have proposed and described the DT framework and application of complex product DMS. Bambura R. et al. [45] constructed a DT framework of the engine manufacturing process consisting of the physical layer, the virtual layer and the information processing layer. The real-time status data collection of the production line and the improvement of production efficiency were realized. Liu SM. et al. [46] proposed an augmented-reality machining process monitoring technology based on a DT machining system, which realized the optimization of the intermediate manufacturing process of complex products based on DT. Mortlock T. et al. [47] argued that the use of DT in manufacturing enterprises can improve the efficiency of the entire manufacturing life cycle and is conducive to realizing the vision of "Industry 4.0". DT can enable enterprises to creatively and efficiently utilize existing manufacturing systems, extract invisible knowledge from experience, achieve more autonomous decision making and control, and improve corporate performance.

At present, reliability data are used by some industrial enterprises for complex product design optimization, manufacturing process optimization and operation and maintenance service reliability and intelligent decision-making optimization. For example, based on the operational reliability data of the Boeing 787, through the design optimization of the power supply system, telecommunication operation technology and new composite materials, etc., the operational reliability of the Boeing 787 has been greatly improved, and its reliability has reached 98.4%. In the meantime, the operation and maintenance costs decreased by 7% compared with the original costs. Bosch Smart Factory in Germany realizes the intelligent optimization and decision making of complex production process by capturing key parameters of key machine components through IoT sensors. Siemens realizes intelligent operation and maintenance and intelligent decision support by analyzing the machine operation status, product quality and manufacturing process reliability data from the operation process of manufacturing enterprise. Therefore, from the existing literature and practical research of industrial enterprises, reliability data bring new directions for the integrated management and intelligent decision making of complex product DMS.

In summary, the traditional PLM method is still very helpful in the FLC management process of complex product DMS. However, by the advancement of advanced digital means, such as digitization, intelligence and visualization in product design, manufacturing and service stages, and reliability data acquisition, analysis and utilization have been further developed. At the same time, the integration of complex product design, manufacturing and service is moving closer and closer in design, manufacturing and service stages. Cross-domain research is becoming more and more prominent, and high integration is becoming more and more important. Therefore, based on reliability data, the research on the integrated cross-domain reliability of complex product DMS has become an urgent problem to be solved to improve the performance and reliability of complex products.

## 3. An Integrated Framework of Complex Product Design Manufacturing and Service Based on Reliability Data

In this paper, we constructed an integrated framework of complex product DMS based on a reliability-data-driven model, as shown in Figure 1, which consists of four parts: complex product DMS integration reliability data acquisition, processing, optimization and innovation application.

As shown in Figure 1, in the integrated reliability data acquisition layer of complex product DMS, it mainly acquires multi-source heterogeneous data, such as product design experiments, product manufacturing and service operation and maintenance. This is performed through 5G, IoT, high-precision sensor, quality management system (QMS), intelligent manufacturing system, manufacturing execution system (MES), etc., to obtain reliability data, such as quality, process, status of complex product DMS process. Through digital processing methods, such as Python, cloud storage and cloud computing, we can carry out data pre-processing, data integration, data analysis and data interpretation on the structured data, semi-structured data and unstructured data in the DMS stage. Data cleaning, transformation, mining and visualization can effectively obtain key reliability data in the process of complex product DMS. This is achieved through the application of reliability data key feature analysis, correlation analysis, deep learning and optimization, clustering analysis and intelligent decision making to achieve complex product design reliability and optimization, manufacturing reliability and process optimization, O&MS and intelligent decision-making optimization. Eventually, the integrated innovative application of DMS, such as innovative design of complex products, manufacturing process optimization and intelligent operation and maintenance and service enhancement based on reliability data, will be realized.

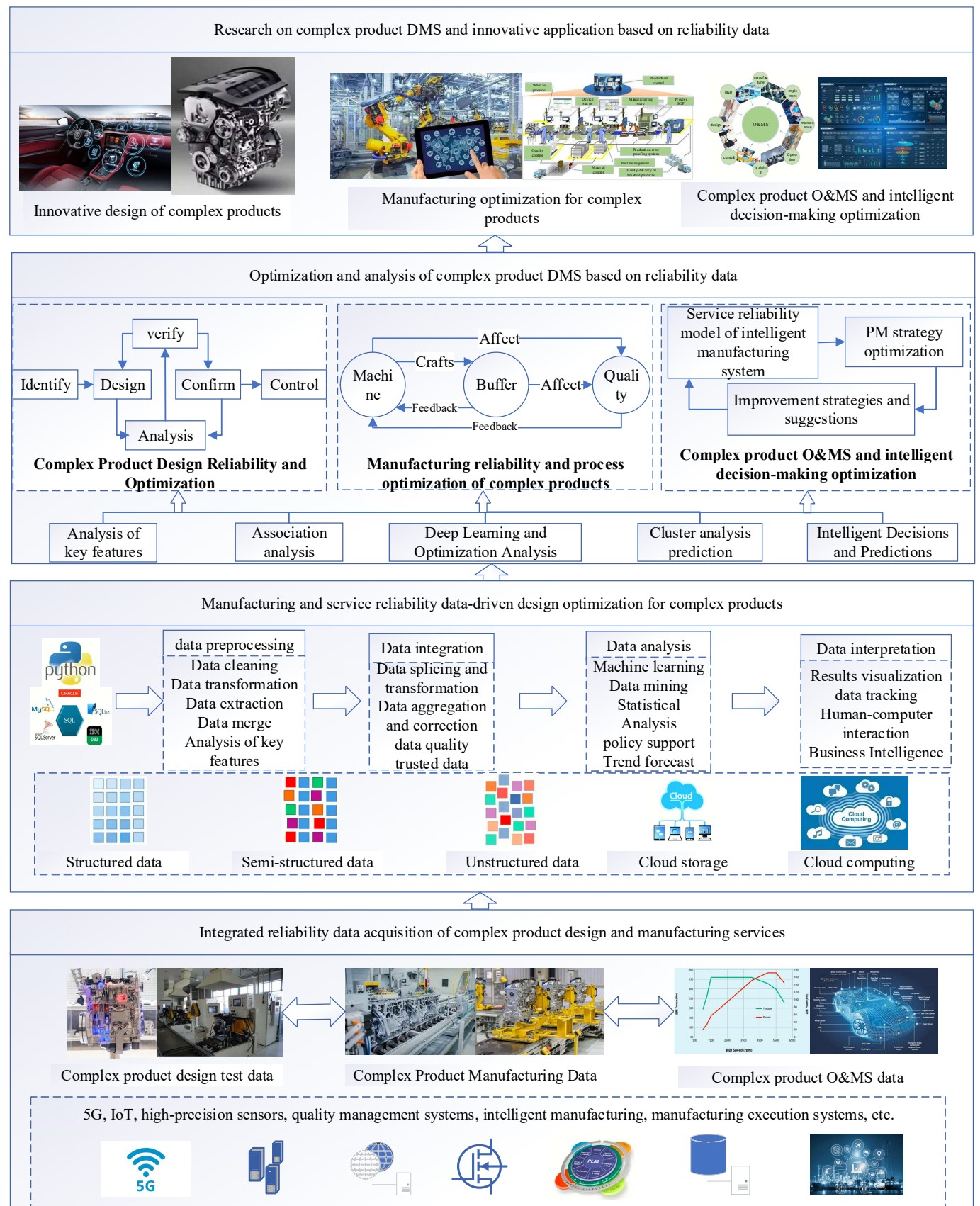

**Figure 1.** An integrated framework of complex product DMS.

### 3.1. Complex Product Design Optimization Driven by Manufacturing and Service Reliability Data

In the past, the product design concept was "test-analyze-and-fix (TAAF)", in which the problems of product design reliability were found only through the test phase [48]. However, with the increasingly complex digital and intelligent competition, the traditional TAAF method is no longer applicable to the complex product design process due to the influence factors, such as design cycle, design and manufacturing costs, maintenance and operation costs, and digital transformation of enterprises, especially for the integrated process of DMS process of complex products. Therefore, reliability should be designed into the product through a more scientific approach. A well-designed finished product may be unreliable in use due to poor production quality. How can we use design analysis, mathematical statistics and analysis, failure mode, effects and criticality analysis (FMECA), deep learning, product usage and maintenance data, lifetime information, critical failure information, intelligent decision making and service, etc., to discover the variation in the manufacturing process and service process? Additionally, the feedback of the corresponding information to the product design engineers, based on manufacturing and service reliability-data-driven optimization of complex product design, becomes a key technical route to solve the reliability of complex product design in the context of intelligent manufacturing.

As shown in Figure 2, the research on reliability-data-driven design optimization of manufacturing and service reliability of complex product mainly includes complex product manufacturing and service reliability data acquisition and database construction, design reliability optimization and innovative applications. In terms of manufacturing and service reliability data acquisition and database construction, first of all, it is necessary to obtain manufacturing and service reliability data, such as manufacturing reliability data represented by product quality reliability data, machine operation state reliability data, process reliability data, etc., and service reliability data represented by critical failure reliability data, life and degradation reliability data, use and maintenance reliability data of complex products, etc. Then, through the database, model base, knowledge base and association rules and mapping rules between manufacturing reliability data and service reliability data, reliability data support is provided for the optimization of complex product reliability design.

The area of complex product design reliability optimization is mainly based on the quality function deployment (QFD) model to identify the influencing factors and control methods that affect the design and product to meet customer needs, based on load strength analysis (LSA) of key features' design parameters and safety margins of manufacturing deviations of complex products and based on DT, performance requirement analysis, design verification and continuous iteration of the key features of complex products. Finally, the manufacturing reliability data and service reliability data are integrated into the general process of complex product reliability design and realize digital design, integrated product development (IPD), green design, DfX, design for reliability (DfR) based on the closed-loop reliability design model driven by dynamic manufacturing reliability data and service reliability data.

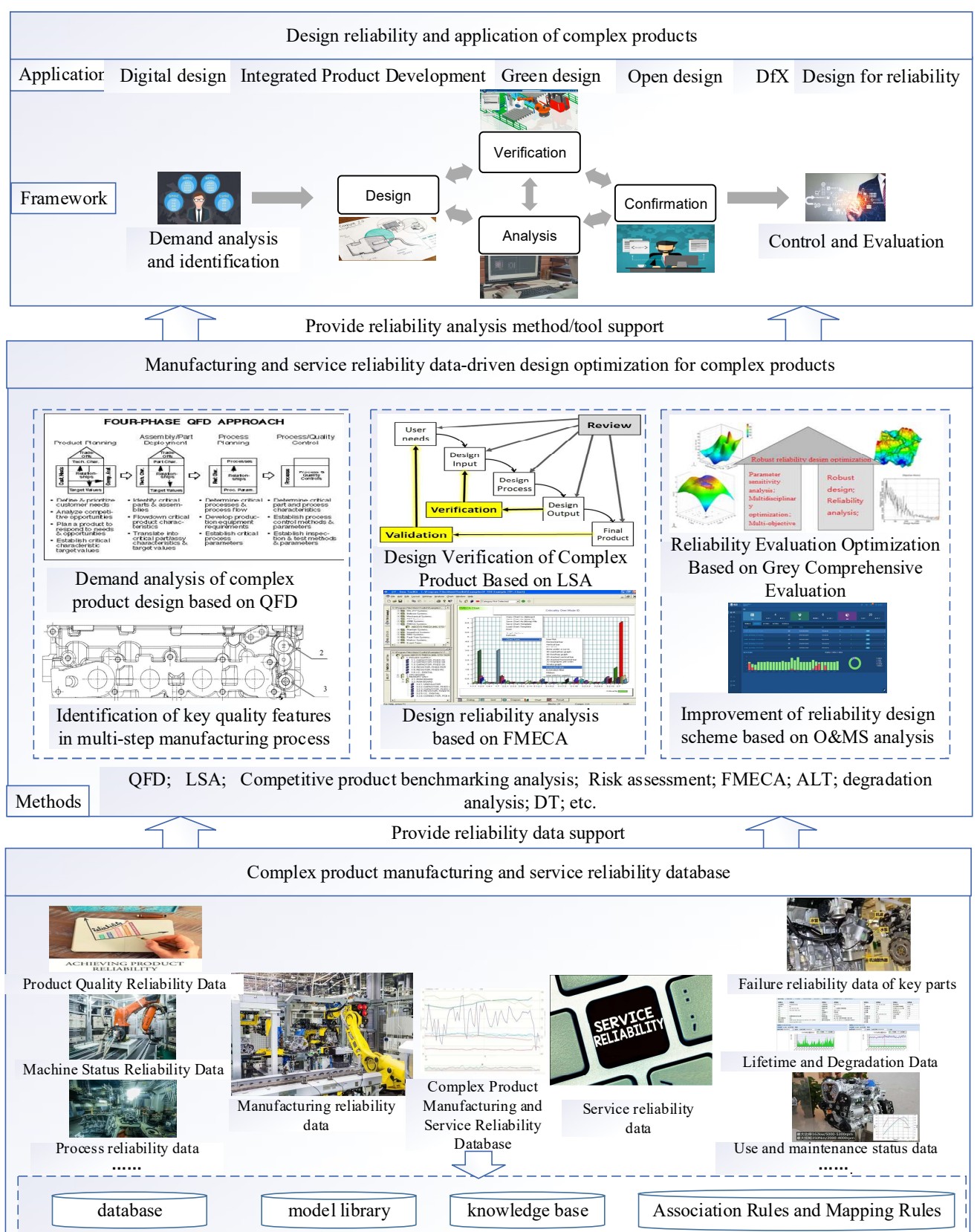

**Figure 2.** Research framework of complex product design reliability optimization driven by manufacturing and service reliability data.

### 3.1.1. Complex Product Requirement Analysis and Key Design Feature Identification

Complex product designers deeply understand and analyze customer needs in the early stage of design, identify key design features of complex product reliability design and fully understand the innovative application of reliability design in all aspects of life cycle management and cross-domain, such as complex product design, manufacturing and O&MS, and intelligent decision making, etc. It is of great significance to maximize the satisfaction of customer needs, shorten the design cycle, improve the reliability of the manufacturing process and reduce service and maintenance costs. Traditional demand analysis methods, such as pre-product research, competitor analysis, project risk analysis and other methods, can still locate consumer demand characteristics. However, with the widespread application of the new generation of information technologies, through Python, deep learning and other methods, it can effectively collect and analyze complex product professional websites and user big data analysis, through high-precision sensors, IoT, etc., to collect dynamic data of manufacturing and O&MS and processes, reliability data of complex products in use and maintenance, and reliability data of life and degradation. It is possible to further mine and identify key design features that affect complex product design.

Based on QFD to identify the influencing factors that affect the design and product to meet customer needs and the key quality characteristics of the integrated process of complex product DMS, it systematically and effectively presents the focus of the reliability design process and the methods to guarantee the reliability design control, which provides methodological support to the market, design, production, reliability and product quality managers to realize the analysis of complex product requirements and identify the key characteristics [49]. Based on the application of big data and other applications in the manufacturing process, the effective identification of key quality features that affect the function of complex products is a key link in the quality supervision and improvement of the intelligent manufacturing process. This is achieved through online collection of data, such as key design features' manufacturing process characteristics of multi-process manufacturing process, intelligent operation of key design features and life state changes in the service process. This is achieved by using machine-learning and intelligent decision-making methods to establish a mapping relationship model between design parameters and state data in the manufacturing and service processes, etc., thereby realizing the reliability design and optimization of key features of complex products with closed-loop feedback mechanism [50].

### 3.1.2. Complex Product Reliability Design and Optimization Process

The new generation of information technology continues to penetrate the field of complex product reliability design, which brings opportunities and challenges to the design reliability of complex products. Design reliability determines the structure, performance and service satisfaction of complex products. More so, it is a key factor that directly affects the competitiveness of complex products in the global scope. As a predictive reliability analysis technology, FMECA is widely used to improve the reliability and safety of complex product design, manufacturing and use phases, and it can also make preventive maintenance decisions for key features of complex products, etc. Therefore, FMECA may be the most reliable and effective reliability analysis method, and its principle is to analyze the failure modes of each key design feature of complex products and, in turn, determine the impact of each failure mode on the overall performance of complex products [51]. In addition, LSA is a method for ensuring that all load and strength cases are considered when designing key features of complex products. It is mainly used in the early stage of complex product design, and with the activities of complex product design, manufacturing and operation and maintenance service data, it can continue in most of the complex product life cycle. The contents of LSA mainly include evaluating the safety margin of the inherent reliability of complex products, analyzing and identifying the realization path of reducing the strength of complex products, finding out the most unfavorable load and strength

data of complex products and their changing trends, especially the key design of complex products' features, etc. [52].

### 3.1.3. Complex Product Reliability Design Control and Evaluation

In order to successfully realize the reliability design of the whole process of complex products and effectively introduce the concept of design reliability into the whole process of complex product design and manufacturing services, it will be helpful to promote the design process of complex products, ensure design quality and reduce design flaws (e.g., structural calculation errors, incorrect selection of equipment materials, radical design, neglect of human reliability, etc.). Therefore, it is necessary to introduce design reliability maturity and related design reliability control and evaluation work in the FLC of complex products. Design reliability is the basis of manufacturing reliability and service reliability of complex products. Combined with the reliability data of manufacturing and operation and maintenance of complex products, a multi-factor reliability design scheme improvement method is constructed from multiple factors, such as manufacturing state, operation state and service state of key design features of complex products, and the factors affecting the reliability of complex product design and the continuous innovation path to improve the reliability design to realize the reliability design of complex products are comprehensively elaborated [53]. On this basis, a tool set can be constructed to control and evaluate the reliability design of complex products using gray comprehensive evaluation, gray fuzzy theory, multi-factor analysis, particle swarm algorithm and high-acceleration stress screening, so as to meet customer needs of innovative design solutions for complex products with high reliability [54].

### 3.2. Complex Product Intelligent Manufacturing Process Optimization Driven by Real-Time Reliability Data

Manufacturing is the key link of complex products, and it is particularly important to optimize the intelligent manufacturing process of complex products based on real-time reliability data. By collecting and analyzing the operation and maintenance inspection, diagnosis and maintenance data generated by key machinery and equipment in the manufacturing process, the design data of key product components and the manufacturing quality, system reliability, etc., for integrated analysis and application, intelligent detection of machine equipment status and buffer management in the manufacturing process are realized. This is performed through interactive application with complex product quality improvement, so as to achieve optimization and innovative application in complex product intelligent manufacturing process based on real-time reliability-data-driven and machine-buffer-quality-reliability model.

As shown in Figure 3, the intelligent manufacturing process optimization of complex products driven by real-time reliability data mainly includes three parts: the acquisition of real-time reliability data in the manufacturing process; the optimization of intelligent manufacturing process driven by reliability data; and the innovation and application of complex product manufacturing reliability. Firstly, it is necessary to obtain design data related to manufacturing reliability (such as functional data, BOM structure and design deviation standards, etc.), manufacturing data (such as process product quality data, inspection data and manufacturing condition monitoring data, etc.) and operation and maintenance service data (such as key component fault detection and diagnosis data, etc.). Additionally, it is necessary to form a real-time, multi-source, heterogeneous database of complex product manufacturing process and corresponding association rules and mapping rules. Secondly, based on the reliability data collected in real time, through deep learning, support vector machines, Monte Carlo, artificial neural network stochastic-flow manufacturing network (AN-SFMN) and quality-reliability (Q-R) methods are used to optimize the analysis and intelligent decision making of the intelligent manufacturing process of complex products. Finally, the machine-buffer-quality-reliability (MBQR) framework is used for the evaluation and innovative application of manufacturing reliability of complex products.

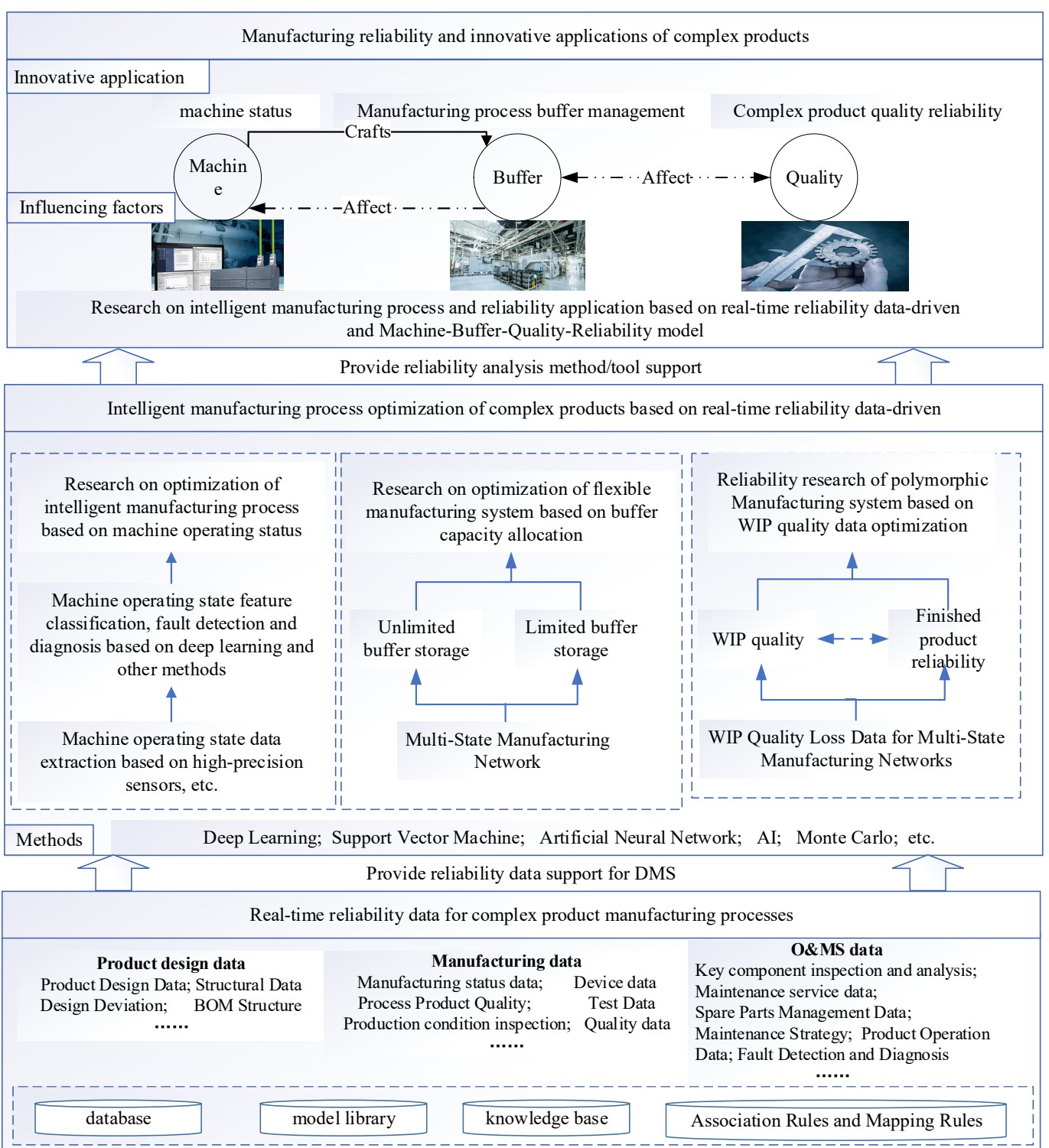

**Figure 3.** Framework for optimization of complex product intelligent manufacturing process based on real-time reliability data drive.

3.2.1. Research on Reliability Optimization of Manufacturing Process Based on Machine STATE

The development of the Industrial Internet of Things (IIoT) enables intelligent manufacturing processes and intelligent control of complex products. Cloud-based remote data collection, intelligent machine interconnection and sensor monitoring technologies provide a new direction for the optimization of intelligent manufacturing processes based

on machine states [55]. In order to better evaluate the technical condition of machine operation, various possible machine state signals, such as failure information of key components, such as bearings, equipment vibration signals and pressure signals, are obtained through high-precision sensors, IIoT, etc. From these feature signals, feature vectors are extracted and counted to identify the machine operation status. Based on multi-sensor data fusion technology, time domain and frequency domain features are extracted from different sensor signals, and deep learning, neural network and other methods are used for feature fusion. The fusion feature vector is used as a machine health indicator for training deep belief network (DBN) for further classification to effectively identify machine operating conditions [56]. Enhancing production plant and manufacturing process control helps improve the overall performance of the complex product manufacturing system. Based on machine-learning, IIoT, high-precision sensors and other technologies and tools, the solution of fault diagnosis, real-time data classification and prediction of machine operating status in the intelligent manufacturing process of complex products has become a priority for the production performance, predictive maintenance and avoidance of energy waste in intelligent manufacturing systems of complex products [57]. Based on reliability-data-driven methods, extracting and processing the failure characteristic signals related to machine damage, degradation and methods to identify degradation patterns related to technical conditions are becoming one of the main directions in the research of reliability optimization of manufacturing processes based on machine states [58].

### 3.2.2. Research on Reliability Optimization of Manufacturing Process Based on Buffer

The polymorphism of buffers, machines and system structures allows the manufacturing system to be flexible and can ensure that the manufacturing system may still not fail completely after a particular machine fails. The buffer capacity has a significant impact on the machine state, and the reasonable allocation of the buffer can effectively improve the average throughput of the manufacturing system and reduce the impact of material interruption on the manufacturing process, which is important in the design of manufacturing system reliability [59]. The Toyota production system and lean manufacturing are widely used in industries such as automobile manufacturing. Manufacturing systems continue to reduce inventory levels and allocate less space for work in progress (as well as raw materials and finished products, etc.), and limited capacity buffers are becoming a trend in manufacturing systems [60]. Multi-state manufacturing system (MMS) has become the main form of the current complex product manufacturing system. The reliability of MMS is expressed as the possibility that all workstations provide enough and sufficient capacity to meet specific needs and the possibility that all buffers do not run out of storage. Further, considering the rework, defect rate and joint buffering, buffer capacity allocation, minimizing allocation cost and evaluating the reliability optimization of manufacturing systems with buffers under maintainable conditions have become the focus of research on reliability optimization of multi-state manufacturing systems [61,62].

### 3.2.3. Research on Reliability Optimization of Manufacturing Process Based on WIP Quality

Quality deviation and production rhythm are the two main characteristics of quality-reliability dependency (Q-R). Multi-state reliability models and dependency theory are often used to characterize the polymorphism of manufacturing systems in the process of degradation, but most studies on MMS dependency ignore the dynamic characteristics of manufacturing systems [63]. Work in process (WIP) is the main output element of the manufacturing process. The quality of WIP is directly related to the operating state of the manufacturing system. For the intelligent manufacturing system process of complex products, there are many deviations (e.g., machining dimensions, surface accuracy, work-in-process quality, machine running status, etc.) in the operating characteristics of the manufacturing system, which are harmful to the operating characteristics. The coupling effect leads to the change of WIP quality and ultimately affects the decline of finished

product reliability, which is reflected as the loss of product manufacturing reliability [64]. Efficient production process design can reduce WIP quality loss, rework, errors and failures, thereby reducing manufacturing system costs and improving the quality of the final product [65]. Based on the correlation between the production task execution state, production equipment degradation state and production product quality state, considering the manufacturing system WIP quality and reliability can optimize the MMS task reliability assessment method and help the development of reliability prediction and health management technology of the intelligent manufacturing system [66].

### 3.3. Complex Product Operation and Maintenance Services Optimization Based on Multi-Source Heterogeneous Reliability-Data-Driven Model

The widespread application of information technology, such as IoT and deep learning, makes it possible to accumulate and deeply mine real-time, massive multi-source heterogeneous data in the FLC of product DMS. Additionally, its effective use can accurately and efficiently improve the reliability of products and O&MS and reduce the cost of operation and maintenance services costs.

As shown in Figure 4, the research on optimization and innovative application of complex product O&MS based on multi-source heterogeneous reliability data drive mainly includes three parts: multi-source heterogeneous reliability data acquisition of manufacturing O&MS, reliability-data-driven optimization of complex product manufacturing operation and maintenance service and innovative application of complex product manufacturing operation and maintenance service. Firstly, for the data islands between DMS, service fragmentation, DMS operation and maintenance disconnection, etc., it is necessary to form operation and maintenance service data interaction rules, cycle iteration and cross-domain closed-loop feedback mechanism, which will be helpful for improving the efficiency, reliability and quality of O&MS. Secondly, through deep learning, artificial neural network, artificial intelligence and reliability analysis and other methods, it is necessary to realize the reliability-data-driven complex product fault detection and diagnosis, preventive maintenance strategy and spare parts management and optimization research. Finally, the innovative applications of complex products in condition detection and feature extraction, intelligent online detection, diagnosis and prediction, and intelligent maintenance decisions and activities are realized through methods such as association and clustering, DT, machine learning and structural equation modeling, and so on.

#### 3.3.1. Research on Fault Detection and Diagnosis of Complex Products Based on Reliability Data Drive

Manufacturing process safety, manufacturing system reliability and product quality of complex products are becoming more and more important in modern industries. In modern digital manufacturing, 79.6% of machine downtime is caused by mechanical failures, and how to mine sensitive fault characteristics from the complex and diverse data signals of long-term mechanical operation and perform fault detection and diagnosis on them becomes the key to the reliability of O&MS [67]. Fault detection and diagnosis (FDD) can identify the root causes of observed system failures, which are crucial to eliminating hidden dangers in manufacturing systems. Timely and accurate FDD of complex product manufacturing processes and systems can effectively improve their reliability and safety. This is different from the traditional FDD based on feature extraction, feature selection and classification and packaging into different modules. It is based on intelligent technologies, such as wireless sensor networks, IoT and DT, to obtain vibration signals, time, frequency and spatial domain data, by using deep learning, neural networks, genetic algorithms, particle swarm algorithms and principal component analysis to achieve fast, efficient and accurate identification of sensitive features based on data and knowledge-based drive and extraction, optimization and classification screening of sensitive features, etc. [68]. Although deep-learning models are widely used in reliability-data-driven FDD methods due to their automatic feature learning capabilities, these models still need to be trained based on historical sensor data, which makes it difficult to meet the real-time requirement of

online fault detection and diagnosis applications. Based on migration learning, knowledge can be learned from the source domain to achieve efficient and reliable solutions to different but similar problems in the target domain, so as to provide online fault detection and diagnosis, prediction and preventive maintenance in the integrated process of complex product DMS [69].

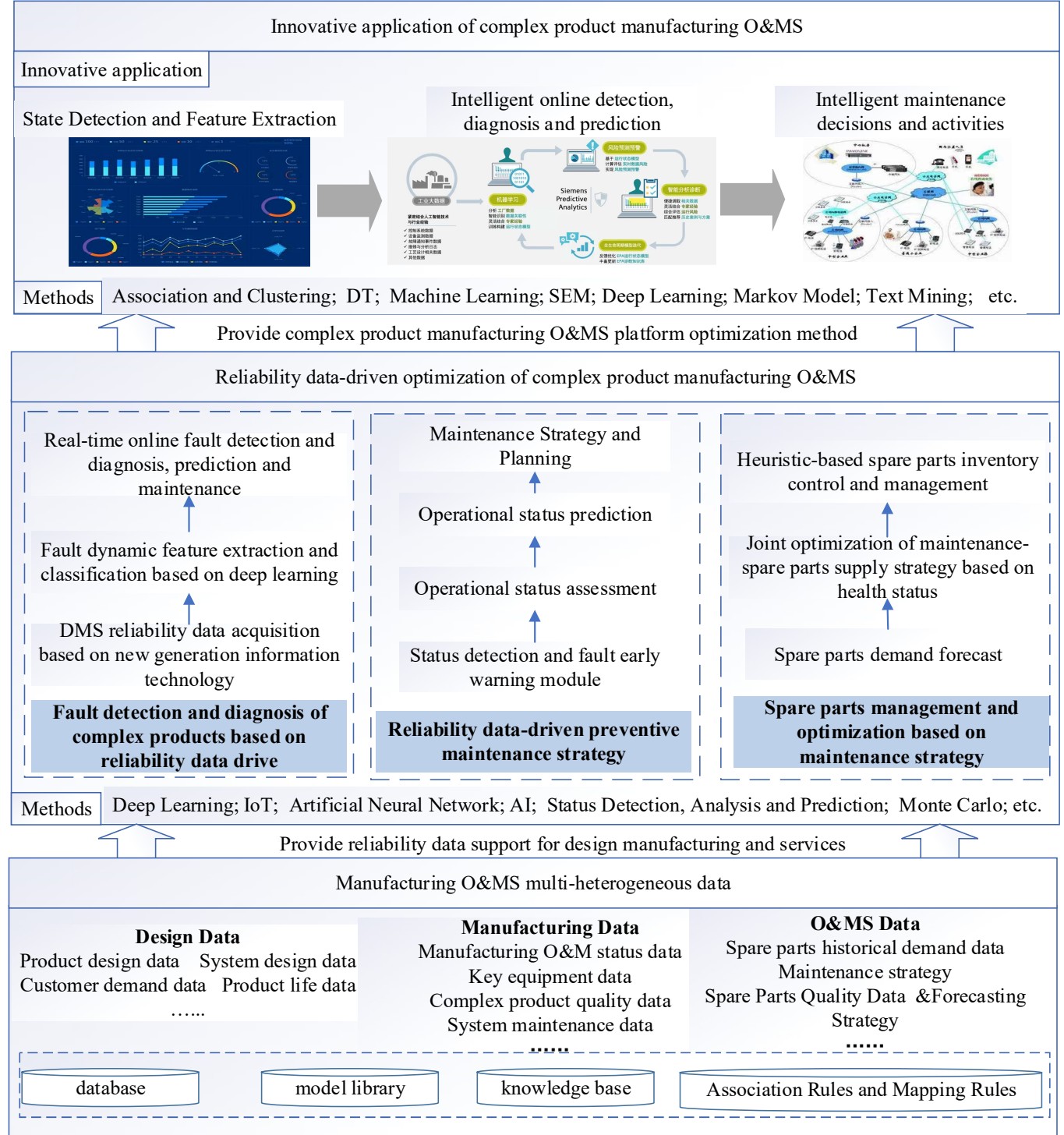

**Figure 4.** Framework for complex product O&MS optimization driven by multi-source heterogeneous reliability data.

### 3.3.2. Research on Preventive Maintenance Strategy Based on Reliability Data Drive

With the application of new technologies, such as smart sensors, Industry 4.0, smart manufacturing, 5G and IoT, in complex product manufacturing processes and operations and maintenance services, automated predictive maintenance is also closely related to robotics. The proliferation of smart sensors has led to an exponential growth in the amount of data extracted from production process. When processed and analyzed, valuable information and knowledge can be identified from manufacturing processes, production systems and equipment, which are used to support increasingly complex system decision making and management through effective analysis and application [70]. Industry 4.0 introduces new paradigms for manufacturing systems, such as digitalization, intelligence, service, networking and platform. Building data-oriented predictive maintenance based on reliability data and IoT technologies achieves a guarantee of the longest uptime of the entire manufacturing chain, reducing production costs while increasing productivity [71]. Predictive maintenance is an effective method to avoid failures and casualties. Taking computer numerical control (CNC) machine tools as an example, as the master machine of industrial manufacturing, if the failure cannot be eliminated in time, it may cause loss of manufacturing accuracy, affect production and reliability of complex products. A hybrid approach based on DT model and DT data-driven model can adequately solve the problem of state diversity and consistency of CNC machine tools during their life cycle, thereby enabling accurate, timely and intelligent predictive maintenance [72].

### 3.3.3. Research on Spare Parts Management and Optimization Based on Maintenance Strategy

Spare parts management is one of the most important aspects of industrial manufacturing systems, especially for highly complex manufacturing processes that require a large stock of spare parts to replace failed components in a timely manner. Spare parts management and maintenance costs are an important part of manufacturing costs, and spare parts management should be considered early in the design and operation phases to reduce downtime in manufacturing systems, and determining the optimal number of spare parts, suppliers and quality of spare parts based on historical data can minimize the total cost of the system [73]. Implementing a secure and trusted spare parts inventory system based on blockchain technology, integrating interplanetary file systems (IPFS) decentralized storage to store and share spare parts data, which is tamper proof, traceable, accessible, immutable, resilient and reliable, provides a new research direction for spare parts inventory management systems to provide reliable ownership tracking of spare parts [74]. In general, manufacturing systems have observable defect information and delayed spare parts availability characteristics; therefore, they have a health-status-oriented drive to achieve joint optimization of preventive maintenance and spare parts inventory management. It can promote the timeliness and robustness of maintenance decisions and reduce the production downtime caused by the delay of spare parts, which is the key to affecting the reliability of intelligent manufacturing systems and complex product O&MS [75]. Additive manufacturing (AM) is considered to be a revolution in the traditional manufacturing technology of spare parts. The application of AM technology in the field of spare parts management can reduce delivery time, waste, energy use, spare parts inventory and improve manufacturing system utilization and reliability of complex product O&MS [76].

## 4. Case Study

### 4.1. Case Description

The engine is the core device that provides the power source for the car and is the heart of the car. The quality of the engine performance determines the reliability, power, stability and environmental protection in the conditions of use of the car. How to achieve short design cycle, low manufacturing cost, high stability of operation and maintenance service, high speed, light weight, low noise, high thermal efficiency and excellent emission perfor-

mance, etc., has become the key for enterprises to continuously pursue the improvement of the overall reliability of engine product DMS.

The GS6 series engine is a high-performance, low-fuel-consumption, all-aluminum gasoline engine independently developed by the SAIC Group, which took 4 years to build and has completely independent intellectual property rights. The GS61 has a maximum power of 138 kw, a maximum torque of 300 Nm, a maximum effective thermal efficiency of 40% and a comprehensive fuel consumption of 5.4 L/100 km. The GS61 engine has 62 patented technologies and applies a series of key technologies, such as in-cylinder direct injection, 350 bar high-pressure fuel injection system, cylinder head and cylinder block cooling, electronic water pump, variable-section turbocharging, continuously variable displacement oil pump, water-cooled intercooling, low-pressure exhaust gas recirculation, rapid phase adjustment variable valve timing (VVT) and variable valve lift, and so on. Through nearly 5 million kilometers of vehicle testing and more than 55,000 h of engine bench testing, the GS6 product R&D and design cycle were shortened from 55 months to 48 months. The manufacturing system average trouble-free time and other reliability indicators were significantly improved, and the O&MS and other intelligent decision-making and dynamic optimizations were realized. At the same time, some key indicators were improved, e.g.: the dynamic response area increased by 39%, fuel consumption decreased by 7%, and idling noise was less than 70 decibels. It successfully realized the core strategy of powertrain for low emissions, low energy consumption and high performance, and its performance, efficiency, quality and reliability are all at the world's top level.

In 2021, global car sales had reached about 81.05 million. As a typical representative of long R&D and design cycles, complex manufacturing processes and high requirements for O&MS, how to realize the integration of engine DMS through cross-domain and collaborative utilization of design reliability data generated in the design process, manufacturing reliability data of complex manufacturing process and reliability data of operation and maintenance service, so as to effectively shorten the engine R&D cycle, realize the optimization of manufacturing process and operation and maintenance service cost reduction goals, are still some of the core issues that need to be solved urgently for current R&D and manufacturing enterprises.

*4.2. A Case Study on the Integration of Engine Design Manufacturing and Services Based on Reliability Data Drive*

The focus is on the integration process of engine assembly DMS through the collection of engine demand characteristics, design information, manufacturing process reliability data and intelligent operation and maintenance service reliability data, based on the cross-domain closed-loop feedback mechanism between design, manufacturing and service. The product development and design cycle are significantly shortened, the overall performance is significantly improved, the mean time between failures and other reliability indicators are significantly improved, the manufacturing cost is significantly reduced, and the intelligence of O&MS is significantly increased, so as to provide consumers with safer and more reliable complex products, as shown in Figure 5.

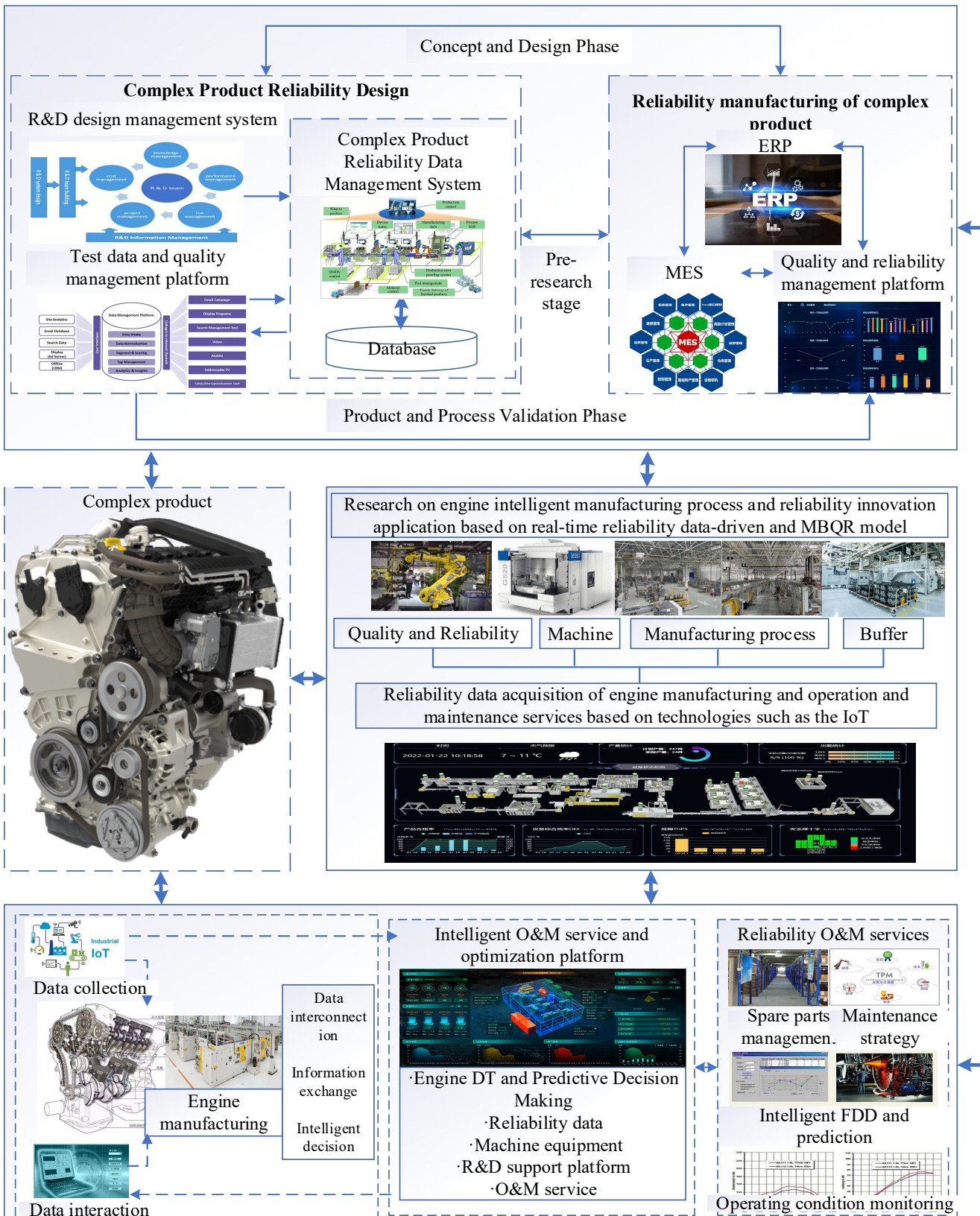

**Figure 5.** Case study of engine DMS integration based on reliability data drive.

### 4.2.1. Design Reliability and Optimization Research

Engine R&D and design is a huge and complex system engineering with deep integration of technology and management, characterized by high investment, high threshold and long cycle, which is characterized by long-term and continuous investment in technology iteration, harsh product manufacturing and operation conditions, high safety standards, high product value and repeated iterations of product development in multiple batches. As a typical complex product with closely related design, manufacturing and service integration, the engine requires a 4–6-year development cycle, and key component development, manufacturing and O&MS may require continuous improvement throughout the life cycle of the complex product, among other characteristics. In the face of an increasingly complex competitive market environment, how to accurately obtain and identify market demand, continuously shorten the R&D and design cycle of complex products, continuously reduce the cross-domain defect transfer from design defects to manufacturing and operation and maintenance service stages, and ensure the reliability of complex products has become one of the main difficulties faced by enterprise R&D and design, manufacturing and operation and maintenance service personnel. The reliability concept is introduced in the pre-research stage, concept and design verification stage, product and process verification stage and manufacturing maturity stage of complex products, etc., to realize the unified standard of design and manufacturing reliability data, traceability of quality and reliability of key components, full participation of manufacturing in front-end design and full consideration of back-end manufacturing and testing in design, so as to realize the continuous improvement of reliability and optimization of complex product design driven by reliability data, as shown in the upper part of Figure 5.

In the pre-research stage, through customer demand analysis, new technology program research, competitive product analysis, manufacturing strategy analysis and product economic and social benefit analysis, the preliminary version of complex product outline diagram, preliminary research reports on manufacturing processes and key components and materials are formed, and design, manufacturing standards are formed. In the concept and design verification stage, according to the market's refined requirements, engineering and manufacturing needs, it is necessary to identify and optimize the bottlenecks in the R&D and manufacturing process, lock the preliminary engineering technology development plan, complete the first prototype trial production and achieve the key performance targets of complex products by verifying and optimizing the design, determine the complete model of complex products, design drawing data, process solutions and engineering bill of material (EBOM) and manufacturing bill of material (MBOM) and product manufacturing strategies, etc. In the product and process verification stage, mass production manufacturing is completed, and the functionality, durability, manufacturability, weight and quality, reliability of key parts and complex products are further verified based on the reliability data of manufacturing and equipment operation and maintenance. It is necessary to provide support for subsequent complex product manufacturing reliability and intelligent O&MS, as shown in Figure 6.

### 4.2.2. Manufacturing Reliability and Optimization Research

Machining is the main core process of the engine manufacturing process. The GS61 series engines use intelligent machines provided by world-class equipment suppliers to complete the complex machining process. Its main core processing equipment comes from Germany, Spain, Italy and China and other world-class powertrain processing equipment manufacturers, such as: Germany—GROB high-precision high-speed dual-spindle machining center; Germany—ELWEMA flexible robot; Spain—ETXE-TAR flexible machining center; and other advanced intelligent processing equipment. The manufacturing process adopts advanced engine manufacturing equipment and technologies such as fully automatic multi-axis CNC machining center, flexible high-pressure parts cleaning, high-pressure seal leak detection and fuel helium inspection, engine hot test, etc., matched with RFID wireless tracking technology and intelligent whole-line monitoring information sys-

tem to meet the real-time application of manufacturing real-time information and product quality information, providing reliable in-process, over-finished and final products. The manufacturing process is continuously optimized and supported, as shown in the middle of Figure 5.

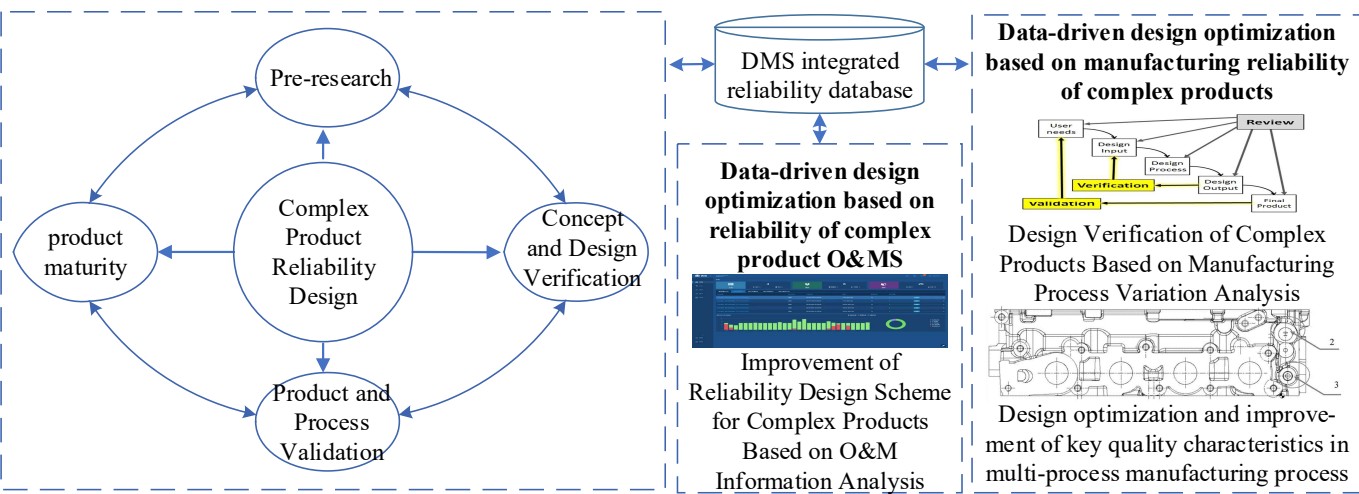

**Figure 6.** Research on reliability design optimization of complex products based on integrated reliability data of design and manufacturing services.

This is based on the IoT, 5G, DT and other new-generation information technologies, through MES, warehouse management system (WMS), QMS, radio frequency identification (RFID) and other means to achieve real-time access and acquire machine operating state data, manufacturing process data, in-process quality data, product quality and reliability data and intelligent decision making. As one of the main components of the engine, the engine block links the engine crankshaft connecting the rod mechanism, gas distribution mechanism, oil supply and cooling into a unified whole, and the engine block processing process and quality reliability have a great impact on the quality and reliability of the whole engine. Taking the engine block production process as an example, through the use of German GROB and other high-precision high-speed dual-spindle machining equipment for milling the cylinder block surface, drilling, reaming, twisting, boring and pinning of various hole systems, the cylinder block blank parts are processed into semi-finished and over-finished products of various processes. After that, non-destructive testing of processing size, hardness, strength and surface flatness is carried out by non-destructive testing technology. Through the air tightness detection test, the cylinder hole penetration and air tightness are intelligently detected. Predictive maintenance of machining equipment, such as tools, controls the machining tool speed, feed, machining depth and width, etc. The quality and reliability of the processed parts are controlled through random sampling and testing. The reliability of the manufacturing system is ensured by rational planning of machining processes and bottleneck manufacturing processes through finished product management. As a result, the finished cylinder body that meets the relevant parameters and quality requirements, such as design strength, stiffness and tightness, is obtained, as shown in Figure 7.

### 4.2.3. Operation and Maintenance Service Reliability and Optimization Research

The design, manufacturing and O&MS of complex products emphasize the real-time collection and processing of equipment data, manufacturing process data and operation and maintenance data, the realization of database, model library and knowledge base and data association rules and mapping rules construction, and the interconnection and interaction of design parameters with manufacturing data and operation and maintenance data through deep learning and artificial intelligence, so as to realize intelligent decision-making process. Through the intelligent operation and maintenance service and optimization plat-

form, enterprises can realize reliability data acquisition, machine and equipment operation status and life data collection, equipment operation and maintenance service data, etc., to interact with R&D support platform in a timely manner. It is necessary to provide auxiliary decision making and continuous optimization for enterprises to realize intelligent equipment detection, diagnosis and prediction, production and manufacturing condition detection, spare parts management and maintenance strategy, etc., thus realizing the reliability and optimization of complex product O&MS based on multi-source heterogeneous reliability data, as shown in the lower part of Figure 5.

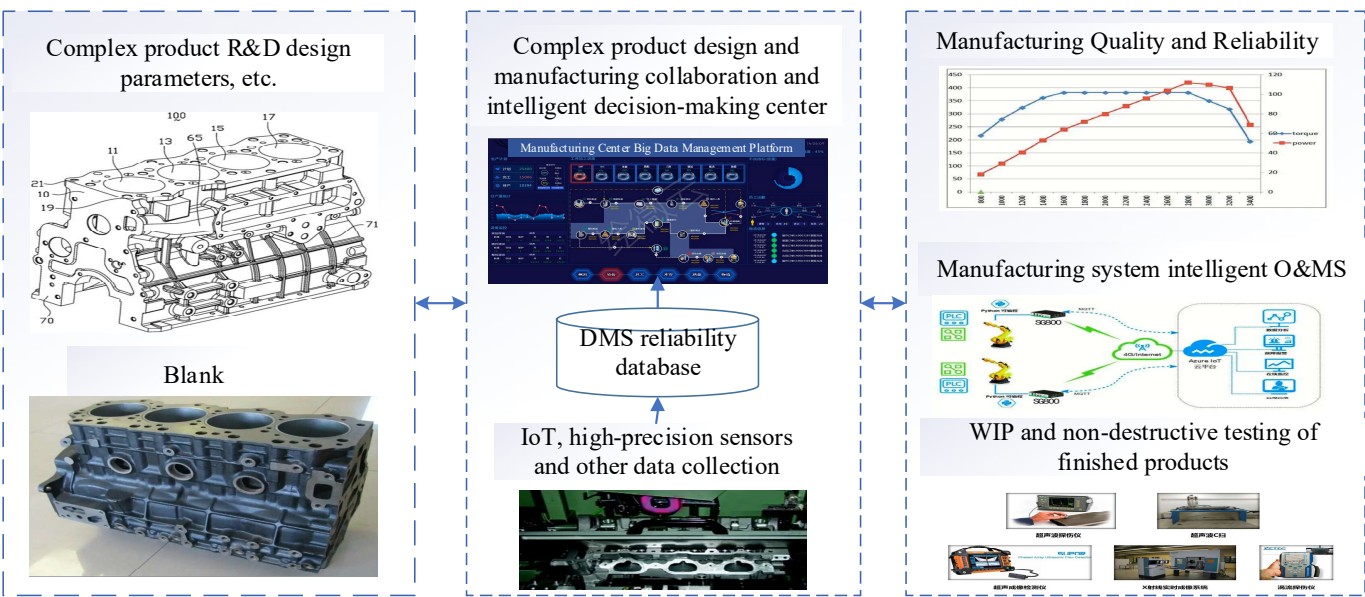

**Figure 7.** Engine block machining process reliability and optimization.

Spare parts management, as the main link of the intelligent manufacturing process of complex products, is of great significance for intelligent equipment to achieve timely maintenance and ensure manufacturing reliability. The development of unmanned supermarket and IoT technology provides new development ideas for unmanned spare parts warehouse management. The use of RFID, computer software and hardware induction control technology, etc., is necessary to achieve the whole process of monitoring, data interoperability and real-time update of spare parts in stock. Real-time inventory query, point-to-point material collection and other business scenarios are realized through UHF RFID chip millisecond-level-induction automatic production billing and scanning system automatic settlement technology. This is performed by using the ERP system to realize convenient operation and management modes, such as unattended spare parts and easy material retrieval. It is possible to carry out paperless inventory counting through tablet computers to achieve green, fast and accurate inventory counting. It is possible to realize lean procurement of inventory spare parts based on historical experience, big data analysis of inventory spare parts, production planning and cost–benefit analysis, etc., as shown in Figure 8.

*4.3. Discussion*

At the same time, in order to further explain the rationality and scientificity of the framework, a questionnaire survey was conducted among R&D engineers, manufacturing engineers and O&MS engineers who participated in the R&D, manufacturing and service of the GS series. The survey content mainly included evaluating whether the framework's R&D and manufacturing services for complex products have been improved and the amount of improvement that still remains.

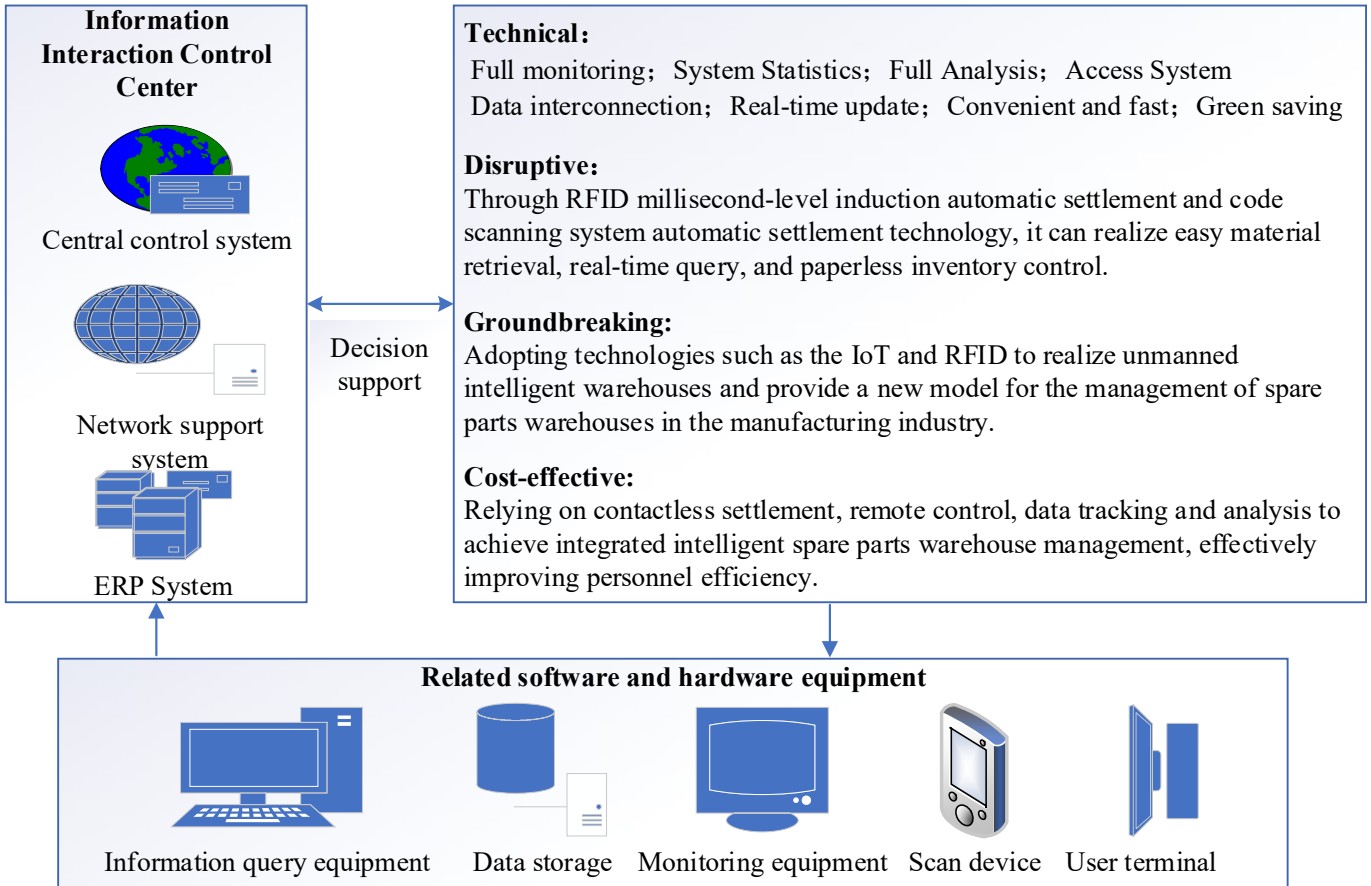

**Figure 8.** Spare parts management for complex product manufacturing system O&MS.

The content of the questionnaire mainly included whether the products have improved in design, manufacturing and O&MS before and after the implementation of the framework. For example: According to your work experience, does the implementation of this framework effectively shorten the development cycle of the GS series? Additionally, choose by how much the R&D cycle has been shortened? At the same time, according to the production report and the research of the production staff, confirm the WIP primary quality pass rate and the primary product reliability (the product is offline and qualified, if there is any behavior such as repair, it is considered that the product is a non-qualified product). As shown in Figure 9.

As shown in Figure 9, the product design cycle was shorter by 12.7%, the primary product reliability (the product is offline and qualified, if there is any behavior such as repair, it is considered that the product is a non-qualified product) increased by about 2%, the maintenance frequency decreased by about 28.6%, and the environmental protection index of the overall product increased by 1%.

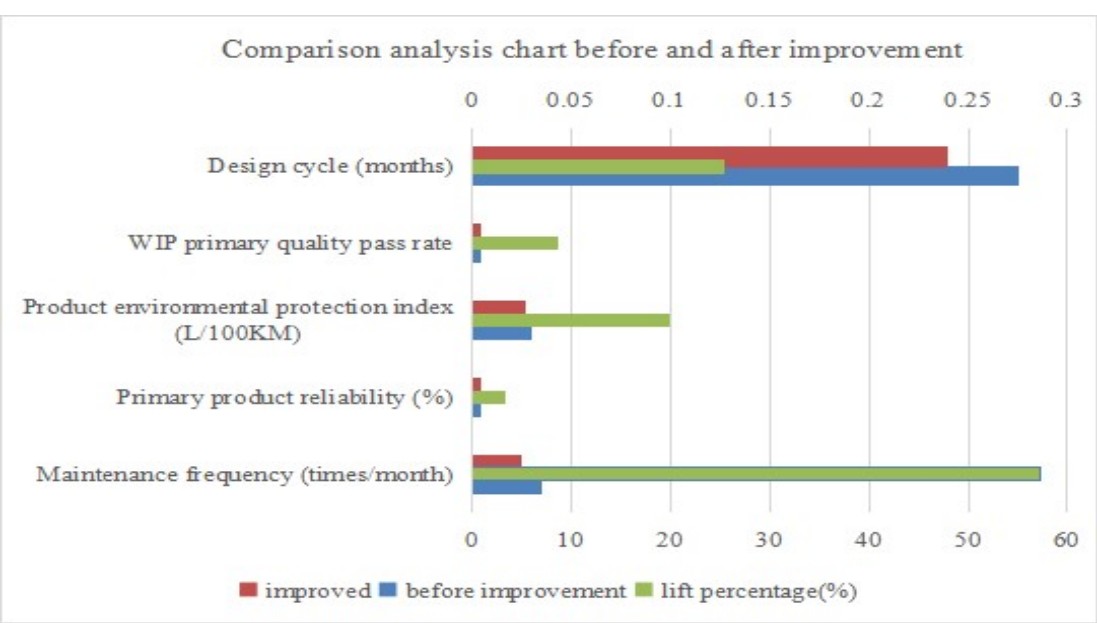

**Figure 9.** Comparison analysis chart before and after improvement.

## 5. Conclusions and Outlook

### 5.1. Conclusions

With the application of IoT, 5G, high-precision sensors and intelligent manufacturing technologies in the FLC of complex product DMS, DMS are increasingly showing the trend of integration. Based on the application of reliability theory in the process of complex product design, manufacturing and service, this paper investigates the technical ways to realize the integration of complex product DMS driven by reliability data, which provides a new method reference to reduce the cross-domain reliability loss in the integration of complex product design manufacturing service and effectively improve the reliability of complex products.

(1) From the current complex product design, manufacturing and service, this paper expounds the problems and deficiencies of complex product design, manufacturing and service at the present stage, and a framework for integrating complex product design, manufacturing and service based on reliability data drive is proposed.

(2) From the application of integrated reliability data of complex product design and manufacturing services, we propose the main technical route for implementing the integrated framework of complex product DMS based on reliability data. Additionally, it includes the optimization of complex product design driven by manufacturing and service reliability data, intelligent manufacturing process optimization of complex products driven by real-time reliability data and optimization of O&MS of complex products driven by multi-source heterogeneous reliability data. Additionally, complex product design reliability and optimization, manufacturing process reliability and optimization and the O&MS and intelligent decision-making reliability and optimization driven by reliability data are then realized.

(3) Taking the DMS integration of engine products as a case study, the design reliability and optimization, manufacturing reliability and optimization, and operation and maintenance service reliability and optimization processes are studied, and the introduction is carried out. The feasibility of the data-driven complex product DMS integration framework proposed in this paper is verified.

### 5.2. Research Deficiencies and Prospects

This paper explores the implementation technology path of reliability theory in the process of complex product DMS integration, and it illustrates the engine DMS process as

an example, which, to some extent, complements the application of reliability theory in the process of complex product life cycle management, especially the integration process of complex product DMS. The R&D and design cycle of complex products is long, the manufacturing process is complex, and there are many uncertain factors in O&MS, and reliability data in the process of DMS present various forms of expression, multiple sources and inconsistent data structures. Therefore, this paper only shows the application potential of reliability-data-driven integration of complex product DMS. In-depth discussions with traditional and emerging methods are still needed in the future to further strengthen the points of this paper. The follow-up of this study still needs to further verify the optimization and innovative application of reliability data in the integration process of complex product DMS from reliability data analysis, which is also the direction and content that can continue to be explored in the future.

**Author Contributions:** Conceptualization, L.F. and J.W.; Methodology, Z.M.; Validation, D.G.; Formal Analysis, Z.M.; Investigation, Z.M. and D.G.; Resources, J.W. and D.G.; Data Curation, Z.M.; Writing—Original Draft Preparation, Z.M.; Writing—Review and Editing, Z.M. and J.W.; Visualization, D.G.; Supervision, L.F. and J.W.; Project Administration, D.G.; Funding Acquisition, J.W. All authors have read and agreed to the published version of the manuscript.

**Funding:** This research was funded by the Innovation Method Fund of China, grant number 2018IM020300; the Innovation Method Fund of China, grant number 2019IM020200; the Joint Funds of the National Natural Science Foundation of China, grant number U1904210-4; the Shanghai Science and Technology Program, grant number 20040501300; the Zhengzhou University Support Program Project for Young Talents and Enterprise Cooperative Innovation Team, grant number 132-32320423; and the General Project of Humanities and Social Science Research for Henan Province's Colleges and Universities, grant number 2023-ZZJH-039.

**Institutional Review Board Statement:** Not applicable.

**Informed Consent Statement:** Not applicable.

**Data Availability Statement:** Not applicable.

**Acknowledgments:** All authors would like to thank the crowd contributors involved in this work.

**Conflicts of Interest:** The authors declare no conflict of interest.

## Abbreviation

| Acronyms | Full Name |
| --- | --- |
| DMS | design manufacturing and service |
| O&MS | operation and maintenance services |
| IoT | internet of things |
| FLC | full life cycle |
| DT | digital twin |
| PLM | product life management |
| PSS | production service system |
| DfX | design for x |
| ADT | accelerated degradation testing |
| CMfg | cloud manufacturing |
| FMECA | failure mode, effects and criticality analysis |
| LSA | load strength analysis |
| QFD | quality function deployment |
| MMS | multistate manufacturing system |
| WIP | work in process |
| FDD | fault detection and diagnosis |
| CNC | computer numerical control |
| AM | additive manufacturing |
| MBQR | machine-buffer-quality-reliability |
| WMS | warehouse management system |

| QMS | quality management system |
| RFID | radio frequency identification |

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
