# Peer review of "Construction of an Integrated Framework for Complex Product Design Manufacturing and Service Based on Reliability Data"

_machines, doi:10.3390/machines10070555_

Round 1
Reviewer 1 Report
In this article, the authors develop and outline an integrated framework for complex product manufacturing based on reliability data. The paper is very well-presented in general and the figures are very clear and easy to follow. The literature review is thorough and helpful. I recommend acceptance of the paper, with some minor suggestions for improvement outlined below.
1. The abstract is not in the correct from for an MDPI journal
2. There are so many technical terms and acronyms used in the papers that it becomes hard to follow in some places. I recommend the addition of a nomenclature table at the beginning of the paper.
3. The length of paragraphs in some parts of the paper are not consistent, which makes the paper less pleasant to read. I recommend trying to form the paragraphs so they are approximately the same length.
4. Please expand the discussion on digital twins - this has a very interesting connection with the presented work and should be explored further.
5. The in-text references are not all in the same style. I recommend using APA or IEEE style for consistency and ease of use.
6. In the final discussion of the paper, it would be good to contrast some of the older/establishing methods (e.g., TRIZ and classic optimization) and emerging methods (e.g., machine learning, digital twins). One of the things I really like about this article is that it gives a balanced and fair perspective, not pushing one specific method but showing how they are all potentially useful - a deeper discussion contrasting the methods would further strengthen this.
7. I would like to see some additional discussion on sensitivities (i.e., problems caused by wrong or incomplete assumptions) and uncertainties with the proposed method.
Good luck and thank you for a very interesting and informative article.
Reviewer 2 Report
The present work describe a developed framework for complex product design manufacturing and service integration, which is based on reliability data. The framework incorporates several types of optimization, such as, the complex product design, the intelligent manufacturing process, the complex product operation and the maintenance services, based on multi-source heterogeneous reliability data. The collected this way data on the different stages of the PLM then are used for take decisions for intelligent optimization of the manufacturing products. The resulting optimizations in the design, manufacturing and maintenance stages due to the proposed framework are demonstrated trough a case study of a vehicle engine ”design manufacturing and service integration”, at the end of the work.
The work has potential to contribute to the contemporary conceptions and approaches for increasing the industry manufacturing processes efficiency, but (according to me) also suffer with some major drawbacks, which should be taken into account and the paper should be reconsidered carefully and revised in deep. They are as follow:
A. Related to the presented and discussed framework and case study:
1. The authors claim in the lines 719 - 724 that the proposed framework lead to significantly improvements of many of the elements of the manufacturing system and the product, and the Fig. 5 is pointed as evidence for that. Unfortunately, I do not see any specific quantitative data given, which can confirm such statements. Usually, the case studies that I have seen are conducted to evaluate the results "before and after" the implementation of some improvement, conception, etc. My questions here are: on what ground you report the significant improvements, what indicators are used and what exactly are the degrees of improvements achieved?
2. Similarly, in the section 4.2.2. which is entitled "Manufacturing reliability and optimization research" are given details about what production equipment, information and software technologies are used, but how you reach to the decision to use these equipment and technologies remains unclear. As result, this part of the work seems more like popular essay rather than scholarly work done.
3. The lack of specific data cited leaves the impression that the proposed framework is still largely at the "implementation" stage and the effects of its application in practice are yet to be assessed. Therefore, the proposed framework cannot yet be claimed that it has been verified, as it stated in conclusion No (3).
4. Finally, I think there is one more aspect that needs to be discussed in the present work: Namely, where is the place, and what is the role of human beings involved in the proposed framework? Regardless of the achievements of all the contemporary “smart” devices and technologies involved in the automated collection, classification and processing of all this data from the processes of design, production and product maintenance services, they have not yet reached a level where there is no human involvement in them. So, people still have roles in the whole these processes, but what will be these roles in the present framework?
B. Related to the way the information is presented:
1. The authors keen to use very long sentences (often reached seven to nine lines) - for example, see the lines 703 - 712. Using shorter sentences will focus the audience attention.
2. Another obstacle for the easily understanding of the discussed problems is the frequent repetition of phrases, which are also too long, even in the one and same sentence. For example, "complex product design reliability", "manufacturing process reliability", "maintenance service reliability", etc. It is advisable these terms to be given as abbreviations, similarly to the other used terms, such as R&D, PLM, IoT, etc. This will simplify the used sentences and make more easily understood to the readers.
3. There are some sentences which meaning is lost - for example see rows 96 and 97 - "The further utilization of high-precision sensors, IoT, 5G, etc. in the process of complex product design manufacturing and service integration."..., and what? The authors should be carefully check the text for such as unfinished sentences.
4. Some of the presented figures are with too low resolution and some of their elements are practically unreadable (for example Fig. 5 and 7).
Reviewer 3 Report
The article is current, because the reliability and safety of manufactured products in industry is very important from the point of view of the competitiveness of organizations and just construction of an integrated framework for complex product design manufacturing and service based on reliability data may be important in the future for both manufacturing organizations and customers. My comments on the article are as follows:
1. In the introduction to the article, the authors mention and point to a new strategy such as "Made in China 92 2025". The authors write that this strategy can be used in product development, production and service. Will it be possible to apply this strategy to services and service development?
2. In Chapter 2.1., the autors point out and mention a number of scientific design methods. Are these methods in the design and development phase or are they already actually applied in production systems?
3. The authors write (line 232) that "Pang JH et al. (2020)[27] developed an intelligent product quality analysis and management system based on Rough Set (RS) and Analytic Hierarchy Process (AHP), and the results showed that a data driven condition monitoring and quality analysis system is an important tool for preventing disasters in complex electromechanical products". Is it possible to use this intelligent product analysis and quality management system based on a coarse file in other products than just electromechanical products?
4. The authors mention the CAE method on line 288. It would also be good to describe what the abbreviations of the letters mean.
5. On line 292, the authors mention the example of a Boeing 787, where they write that "Boeing 787 has been greatly improved and the operation and maintenance costs have been significantly reduced". Would the authors be able to quantify how much it is in %?
6. In Chapter 3.1, the authors write on line 356 that "A good design may cause the overall product reliability to decrease due to the variation in the manufacturing process". I do not agree with this statement, I recommend the authors to think about this sentence and reformulate it.
7. In subchapter 3.1.3, the term design risks. In the article, it would be good to name which are the specific risks?
8. In subchapter 3.2.1, the authors used in the sentence "In order to better evaluate the health condition of machine operation, various possible machine state signals, such as failure information of key components such as bearings, equipment vibration signals and pressure signals, are obtained through high-precision sensors, IIoT, etc." I propose to change the word health condition to technical condition, as this is a technical device. The same goes for the sentence on line 528.
9. In subchapter 3.2.3, the authors write that "For the intelligent manufacturing system process of complex products, there are many deviations in the operating characteristics of the manufacturing system, which are harmful to the operating characteristics". It would be good to add to the article what deviations are involved?
10. On line 679, the authors used the expression environmental protection of the car. Here I would suggest to use another term, namely environmental protection in the conditions of use of the car.
11. I suggest adding a reference to the article in the article: "Bambura, R. et al. Implementation of Digital Twin for Engine Block Manufacturing Processes. Appl. Sci. 2020, 10, 6578. https://doi.org/10.3390/app10186578"
12. Images could be larger because the text in the images is small and difficult to read.
Round 2
Reviewer 2 Report
Ok. I accept the corrections made by authors in the second edition of the paper, and the answers given in the authors' response.
Reviewer 3 Report
Dear authors
Thank you very much for your answers to my questions about the article entitled Construction of an Integrated Framework for Complex Product Design Manufacturing and Service Based on Reliability Data.
Your answers are sufficient, you have improved the article and incorporated all comments into the article. I recommend you to continue your research and I wish you much success in publishing in the subject matter.